Tip dating with fossil sites and stratigraphic sequences

King Benedict benedict.king@naturalis.nl
Rücklin Martin
Naturalis Biodiversity Center , Leiden , Netherlands
López-Antoñanzas Raquel
Electronic publication date: 2020 Jun 26
Publication date: 2020
Volume: 8
Electronic Location ID: e9368
Received 2020 Feb 5; Accepted 2020 May 26
Copyright: ©2020 King and Rücklin
Copyright year: 2020
Copyright holder: King and Rücklin
License: This is an open access article distributed under the terms of the Creative Commons Attribution License, which permits unrestricted use, distribution, reproduction and adaptation in any medium and for any purpose provided that it is properly attributed. For attribution, the original author(s), title, publication source (PeerJ) and either DOI or URL of the article must be cited.
License URL: https://creativecommons.org/licenses/by/4.0/

Keywords: Tip-dating, Fossils, Stratigraphy, Prior, Acanthodians, MOTH, Turin Hill, Devonian, BEAST2

Funding: Dutch Research Council NWO Vidi 864.14.009 This work was supported by Dutch Research Council NWO Vidi grant 864.14.009. The funders had no role in study design, data collection and analysis, decision to publish, or preparation of the manuscript.

==============================
Tip dating, a method of phylogenetic analysis in which fossils are included as terminals and assigned an age, is becoming increasingly widely used in evolutionary studies. Current implementations of tip dating allow fossil ages to be assigned as a point estimate, or incorporate uncertainty through the use of uniform tip age priors. However, the use of tip age priors has the unwanted effect of decoupling the ages of fossils from the same fossil site. Here we introduce a new Markov Chain Monte Carlo (MCMC) proposal, which allows fossils from the same site to have linked ages, while still incorporating uncertainty in the age of the fossil site itself. We also include an extension, allowing fossil sites to be ordered in a stratigraphic column with age bounds applied only to the top and bottom of the sequence. These MCMC proposals are implemented in a new open-source BEAST2 package, palaeo. We test these new proposals on a dataset of early vertebrate fossils, concentrating on the effects on two sites with multiple acanthodian fossil taxa but wide age uncertainty, the Man On The Hill (MOTH) site from northern Canada, and the Turin Hill site from Scotland, both of Lochkovian (Early Devonian) age. The results show an increased precision of age estimates when fossils have linked tip ages compared to when ages are unlinked, and in this example leads to support for a younger age for the MOTH site compared with the Turin Hill site. There is also a minor effect on the tree topology of acanthodians. These new MCMC proposals should be widely applicable to studies that employ tip dating, particularly when the terminals are coded as individual specimens.

Introduction

Tip dating is increasingly used as a method to calibrate molecular phylogenies and to analyse phylogenies with fossil taxa (Gavryushkina et al., 2017; Lee et al., 2014; Ronquist et al., 2012). Central to tip dating are the ages given to the individual fossil taxa (or tips). Often this has been in the form of point estimates, but usually there is some uncertainty regarding the precise age of a fossil. Simulations have shown that better performance is achieved when each fossil is given a uniform tip age prior across the range of uncertainty (Barido-Sottani et al., 2019).

Current implementations of tip dating only allow tip ages to vary independently from each other. This has the undesired effect of separating the ages of fossil taxa from the same site. In reality, it is frequently the case that a fossil site has a wide uncertainty regarding age, but it is known that all the fossils from that site are of approximately the same age. A striking example, from the empirical dataset used in this study, is the so-called “wonder block” from the Man On The Hill (MOTH) site from the Lochkovian (419.2–410.8 million years) of northern Canada (Hanke & Wilson, 2006). This single block contains the acanthodians Obtusacanthus, Brochoadmones and Lupopsyrus (Hanke & Wilson, 2006), but these fossils can be separated by millions of years in a tip-dated analysis in which age uncertainty is dealt with in the typical manner (King et al., 2017).

A second limitation is that a series of fossil sites can often be placed in chronological order, despite the upper and lower bounds for the age uncertainty of these sites overlapping. For example, radiometric dates might be known only for the top and bottom of a geological formation, and fossils known from several layers within this formation. Therefore, although the total range of uncertainty for the age of each fossiliferous layer is the same, it is known in which order the layers occur. In current implementations of tip dating, it would be necessary to either impose arbitrary age bounds to maintain the chronological order of the layers, or to allow the layers to be sampled in the incorrect order.

In this study we introduce new MCMC proposals for the software BEAST2 (Bouckaert et al., 2019), which allow linking of tip ages for fossils from the same site, as well as the ordering of fossil sites within a stratigraphic sequence. We test these proposals on a dataset of early gnathostome fossils (King et al., 2017), focusing on two fossil sites with multiple taxa but wide age uncertainty ranges: the Man On The Hill (MOTH) site from the Lochkovian (Early Devonian) of Canada, and the Turin Hill (or Tillywhandland) site from the Lochkovian of Scotland.

Materials & Methods

Tip date MCMC proposals for fossil sites are implemented with new operators in BEAST2, available in the BEAST2 package palaeo (available for download at https://github.com/king-ben/palaeo). The package includes an R function that generates xml code for these operators from tables of fossil site occurrences.

The first operator, FossilSiteDateRandomWalker, an extension of SampledNodeDateRandomWalker from the sampled ancestors package (Gavryushkina et al., 2014), takes as input the list of taxa and the age bounds for the site. New proposals, consisting of a random age within the upper and lower bounds, are applied to all taxa in the site simultaneously. Each fossil site requires a separate operator. A second operator, RelativeFossilSiteDateRandomWalker, allows the ordering of sites within a stratigraphic sequence, while allowing overlapping upper and lower bounds. The additional inputs are fossil sites that sit above and/or below within the sequence. This operator effectively implements a prior on the relative ages of fossil layers. New proposals for fossil site ages that fall outside bounds defined by the age estimates for the sites occurring immediately below or above within a sequence are immediately rejected (i.e., assigned a prior probability of 0). The acceptable bounds for the age estimates of sites therefore depend on the other sites in the sequence, and will change as the MCMC chain runs (Figs. 1A–1B).

Figure 1 An MCMC proposal enforcing the correct ordering of fossil sites within a sequence, but allowing overlapping uncertainty bounds.

(A–B) The blue lines represented the sampled ages of two fossil sites (light blue: younger site, dark blue: older site) . The range of possible values for new proposals at a particular point in the Markov chain (part A and B represent different points in the chain) depends on the current value of the other site in the sequence. Arrows indicate the possible range of new proposals (proposals outside this range are assigned a prior probability of 0). (C) Implementation of this operator on an empirical dataset leads to non-uniform effective priors on site age (in this case two formations from the Early Devonian of Spitzbergen). Note that colours are plotted with transparency to show overlap.

We tested the performance of these new MCMC proposals on the dataset of King et al. (2017), a dataset that includes autapomorphies, which can be important for tip dating (Matzke & Irmis, 2018). We updated the BEAST2 xml files, first with independent age priors for each fossil, using the SampledNodeDateRandomWalker operator from the sampled ancestors package (Gavryushkina et al., 2014), and second with the new MCMC proposals described here. Analyses used an uncorrelated lognormal clock (Drummond et al., 2006). The prior on clock rate was an exponential with mean 0.003 and offset 0.0016, while the prior on clock standard deviation was an exponential with mean 1. We used the Mkv model (Lewis, 2001) and gamma distributed among-character rate variation (Yang, 1996) with four rate categories; the prior on the shape parameter was a uniform distribution on the range 0–10. The tree prior was a sampled ancestor birth-death model (Gavryushkina et al., 2014), with a lognormal distribution prior (mean in real space 0.14, standard deviation 0.9) on birth rate, an exponential prior (mean 0.1) on death rate and an exponential prior (mean 0.03) on sampling rate. The analyses were run for 200,000,000 generations across four independent runs, with a 10% burn-in, and convergence was confirmed using Tracer (Rambaut et al., 2014) and Rwty (Warren, Geneva & Lanfear, 2017). To test the effect of using a RelativeFossilSiteDateRandomWalker operator, we also ran an analysis sampling from the prior only.

Post-processing of results was performed in R (R Core Team, 2018), utilising the packages ape (Paradis, Claude & Strimmer, 2004), phytools (Revell, 2012) and ggplot2 (Wickham, 2016). Following the recommendations for summarising trees in O’Reilly & Donoghue (2018), 50% majority-rule consensus trees were calculated in the R package ape (Paradis, Claude & Strimmer, 2004) and posterior probabilities were calculated for nodes on this tree in TreeAnnotator 1.10.2 (Suchard et al., 2018). We focused attention on two sites in particular, the Man On The Hill site (MOTH) and Turin Hill. Both have wide age uncertainty (Lochkovian, 419.2–410.8 Million years), but contain several taxa: eight and five acanthodians respectively for this particular analysis. Full results and analysis scripts are available on the github repository (https://github.com/king-ben/palaeo).

Results

We demonstrate the effect of the RelativeFossilSiteDateRandomWalker on taxa from the Devonian Red Bay Group from Spitzbergen, specifically taxa from the older Fraenkelryggen and younger Ben Nevis formations (Fig. 1C). As expected, implementation of this operator led to a non-uniform effective prior on tip ages (Fig. 1C). The effective prior on age of the Fraenkelryggen Formation taxa was concentrated in the older part of the age uncertainty range, while the converse was true for the Ben Nevis Formation taxa. However, values across the entire span were sampled for both.

When estimated independently (i.e., with SampledNodeDateRandomWalker), individual taxa from the same fossil site could show widely variable dates within a single tree from the posterior sample. Across the posterior sample, the age estimates for taxa from the Turin Hill site were spread over an average range of 5.43 million years, while the range for MOTH was 6.55 million years.

Figure 2 Linking the tip ages of fossils from the same site leads to increased precision of age estimates and has minor effects on tree topology.

(A) 95% HPD intervals for individual taxa within two fossil sites (light green, Turin Hill taxa; light orange, MOTH taxa), compared with 95% HPD interval when tip ages within fossil sites are linked (dark green, Turin Hill; dark orange, MOTH). Circles represent median estimates. (B–C) 50% majority rule cladogram (in part) from the analysis with independent tip ages (B) and with linked tip ages for fossil sites (C).

The FossilSiteDateRandomWalker operator resulted in increased precision of site age estimates when compared with estimates for each taxon independently (Fig. 2A). The 95% highest posterior density (HPD) interval spanned 4.79 million years for the MOTH site, whereas the HPD intervals for individual MOTH taxa when given independent ages spanned between 5.53 and 7.92 million years. The HPD interval for the Turin hill site spanned 6.08 million years, compared with between 7.36 and 7.77 million years for the individual taxa.

The analysis with linked ages within each fossil site supported a younger age for the MOTH site compared to Turin Hill. The median age for MOTH was 412.36 million years (HPD 410.80–415.59), whereas the median for Turin Hill was 416.47 million years (HPD 413.06–419.15). MOTH was younger than Turin Hill in 97% of trees from the posterior sample.

Some support a younger age for MOTH than Turin Hill was also present in the analysis with independent dates, although the effect was less strong. The age estimate for the two sites in each sample from the posterior was calculated as the mean of the age estimates for the individual taxa. The median across the posterior sample of this estimate was 414.57 for MOTH and 415.40 for Turin Hill. The mean age for MOTH taxa was younger than the mean for Turin Hill taxa in 76% of trees from the posterior sample. Therefore, use of linked tip dates amplifies support for a younger age for MOTH. Notably the median age estimate for the MOTH site when tip ages were linked was younger than the median age estimate for any of the individual MOTH taxa when tip ages were independent (Fig. 2A). Conversely, the median age estimate for Turin Hill was older than the estimates for any of its individual taxa (Fig. 2A).

Use of linked tip dates had a minor effect on tree topology (Figs. 2B–2C). The 50% majority-rule consensus tree for the analysis with independent tip dates showed the MOTH taxon Cassidiceps resolved as the sister group to a clade consisting of Mesacanthus, Promesacanthus, Cheiracanthus, Homalacanthus and Acanthodes (Fig. 2B). When tip dates within fossil sites are linked, this node collapses into a polytomy (Fig. 2C).

Discussion

The results show that linking tip ages from fossil sites can lead to an increase in the precision of age estimates when compared with analyses allowing independent tip dates. This may have important implications for the use of Bayesian phylogenetic estimation of fossil ages (Drummond & Stadler, 2016), in cases where a fossil site has uncertain dates but multiple taxa. For example, in an analysis estimating the age of fossil sites containing phiomorph rodent fossils, the estimated ages for fossils within a single site were sometimes different (Sallam & Seiffert, 2016). The use of linked tip dates should therefore increase the accuracy and precision of such estimates. In theory, further extensions to these tip date operators could even allow the use of multiple trees (i.e., several groups analysed simultaneously), with dates for fossil sites linked across the trees, to further increase precision.

The age estimates for the MOTH and Turin Hill sites should for now be treated with caution. The younger age estimate for the MOTH site is likely driven by the similarities of some taxa with chondrichthyans (Hanke & Wilson, 2004; Hanke & Wilson, 2010), and others with diplacanthid acanthodians (Hanke & Davis, 2008; Hanke, Davis & Wilson, 2001). The earliest chondrichthyan fossil for which good morphological data is known is Doliodus (Miller, Cloutier & Turner, 2003), of Emsian (late Early Devonian) or early Eifelian (early Middle Devonian) age. Diplacanthid acanthodians are mainly found in the Middle Devonian (Burrow et al., 2016). However, the presumed poor sampling of early chondrichthyans in the fossil record (Coates et al., 2018), combined with the sparsity of morphological characters that can be coded for even the best-preserved acanthodian fossils, means that this result should be considered preliminary.

Linking tip dates can affect the phylogenetic position of fossils. The affected taxon, Cassidiceps, from the MOTH site, has the oldest age estimate of the MOTH acanthodians when tip dates are allowed to vary independently (Fig. 2A). Enforcing all MOTH taxa to have the same age therefore leads to a larger difference in the sampled age for Cassidiceps, leading to increased sampling in a more nested position (Fig. 2C). Relative to other MOTH acanthodians, the morphology of Cassidiceps is relatively poorly known (Gagnier & Wilson, 1996), which is likely to further increase the relative influence of tip age priors on its phylogenetic position.

The effect of the stratigraphic sequence operator on the effective tip date prior (Fig. 1C) is desirable. For example, for the younger fossil layer to be close to the maximum age bound for the stratigraphic sequence (e.g., Fig. 1B) would require both fossil layers to occur very close together, implying highly heterogeneous sedimentation rates. While this is possible, it is appropriate to assign a low prior probability to this scenario. Further refinements to tip age priors could be added, such as applying non-uniform tip age priors (in addition to the operators), based on the relative position of a layer in a sequence and an assumption of uniform sedimentation rates. The combined effect of the tree prior, operators and tip age priors on the effective prior density on tip dates would need to be analysed by sampling from the prior, as for node age calibrations (Heled & Drummond, 2012).

The MCMC proposals presented here are particularly relevant for specimen-level phylogenetic datasets (e.g., Cau, 2017; Tschopp, Mateus & Benson, 2015). Even when phylogenetic datasets are not strictly specimen-based, taxa are often scored based on a single specimen or specimens from a single fossil site (as is the case for the phylogenetic dataset utilised here). Simulations have shown that tip dating works best when fossil terminals are assigned ages based on the specimens from which the morphological data were coded (Püschel et al., 2020). The new methods presented here should therefore be widely applicable to phylogenetic analyses of palaeontological data. This includes the use of “clock-less” tip dating to timescale trees, without the use of morphological data (paleotree R package v. 3.3.25 reference manual; Bapst, 2012). Correctly handling of fossil ages is also of importance for analyses utilising the unresolved fossilised birth-death model (Heath, Huelsenbeck & Stadler, 2014), and there is increased appreciation for the need to include adequate fossil samples in such analyses (O’Reilly & Donoghue, 2020).

We note that these MCMC proposals will not be applicable to all fossil occurrences. Where sedimentation rates are slow, fossil sites can cover millions of years in time; in this case it would be more appropriate to analyse fossils layer by layer using a stratigraphic sequence operator. However, these proposals will be inappropriate when there is considerable reworking of fossils, leading to uncertain relative ages. In cases where morphological data for individual tips are taken from fossils covering a wide stratigraphic range, models that explicitly take stratigraphic ranges (as opposed to uncertainty) into account would be more appropriate (Stadler et al., 2018), although these are not yet implemented in phylogenetic software.

Conclusions

This study introduces new MCMC proposals implemented in BEAST2 designed to deal with stratigraphic age uncertainty of fossils by linking the ages of fossils from the same site, as well as correctly ordering fossil sites within a sequence. When used on an empirical dataset, the use of these new proposals leads to increased precision of site age estimates and minor effects on tree topology. The MCMC proposals presented here should be widely applicable to studies that employ tip dating, particularly for specimen-level datasets.

We thank Remco Bouckaert for advice on BEAST2 and Daniele Silvestro, Eric Gorscak and an anonymous reviewer for comments on the manuscript.

Additional Information and Declarations

Competing Interests

Author Contributions

Data Availability

The authors declare there are no competing interests.

Benedict King conceived and designed the experiments, performed the experiments, analyzed the data, prepared figures and/or tables, authored or reviewed drafts of the paper, and approved the final draft.

Martin Rücklin conceived and designed the experiments, authored or reviewed drafts of the paper, and approved the final draft.

The following information was supplied regarding data availability:

The Beast2 addon package palaeo is available at Github: https://github.com/king-ben/palaeo, together with the analysis files used in this article.

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
