# Peer review of "Tip dating with fossil sites and stratigraphic sequences"

_PeerJ, doi:10.7717/peerj.9368_

## Round 0.1 · original submission · Major Revisions

Dear Dr King,

Your manuscript “Tip-dating with fossil sites and stratigraphic sequences” has now been reviewed. The reviewers have recommended publication after revision. Nevertheless, I am particularly concerned about the accuracy of this method to be used on analyses of fossils alone (see O’Reilly and Donoghue 2020) and I would appreciate if you could discuss this issue in your paper.

Important points raised by the reviewer include concerns about the age of the sites “(1) fossils from the same site are not necessarily of identical ages (2) there are cases in which sedimentation rates are very low resulting in millions of years of evolutionary history being compressed in a few centimeters of rock. (3) sediments might be reworked resulting in unclear relative ages of occurrences within a stratigraphic column.”
Please act on the comments of the referees and mine and revise the manuscript accordingly. When resubmitting your revision you should provide a document indicating how you have responded to each of these points.

Thank you very much for your attention.

Best regards,
Raquel Lopez-Antoñanzas

·

Basic reporting

-

Experimental design

-

Validity of the findings

-

Additional comments

King and Rücklin present a new module for BEAST2 introducing new operators to update tip ages while accounting for the fact that relative ages of fossils might be known. For instance, two fossils from the same site can have a range of temporal uncertainty associated with the dating of the site but can be assumed to be of identical ages. The new operator constrains the proposal for ages of the two extinct tips to be identical, within their stratigraphic range.

I think the paper is very well written and presents an interesting and useful tool. There are however a couple of things that I would like the authors to clarify about their method. While sampling identical ages for two or more tips from the same site is relatively straightforward, It is unclear how tip ages are sampled in the case of ordered stratigraphic ranges (lines 79-84). In particular, because the update of one tip age is not independent of the other tip ages, it is not clear whether the move is reversible and symmetrical and whether there should be a correction in the form of a Hastings’ ratio. One simple solution would be to

1) draw n ages (for n tips) from a uniform distribution bounded at the stratigraphic boundaries of the section
2) sort them
3) assign them to the tips, sorted by relative age

This move should ensure that the proposal is symmetric and reversible. Maybe this is what the authors have already implemented, in any case I think this point should be clarified in the manuscript.

While I agree with the authors that the possibility of constraining fossil ages based on their sampling site is generally a good idea (in fact my co-authors and I have recently introduced a similar feature in a non-phylogenetic context: doi.org/10.1017/pab.2019.23), I think the authors should briefly discuss the fact that two fossils from same site are not necessarily of identical ages. There are cases in which sedimentation rates are very low resulting in millions of years of evolutionary history being compressed in a few centimeters of rock. Worse even, sediments might be reworked resulting in unclear relative ages of occurrences within a stratigraphic column. I think briefly mentioning these issues a would be a useful cautionary note.

Minor point: Change “I” for “we” in the Abstract.

I hope you will find these comments useful.
Best regards,
Daniele Silvestro

Reviewer 2 ·

Basic reporting

Although reference is made to a github site, it is not clear what is actually available there.

Also, on line 93 the authors refer the reader to methods as used in King et al. 2017. It is important that the authors report the details of the analysis
as they performed it in this paper (or in Support Mater) and not just refer the reader to another paper.

Experimental design

Methods in King et al. 2017 are not sufficiently described. See above under 1.

Validity of the findings

Conclusions with respect to the test data are perhaps overstated. The authors claim that linking tip-date priors is a good idea, and they provide a way to do this. So far, I agree.

However, their empirical example does not convince me that this linking of priors actually has a significant effect, or is even an improvement.

Brnach support values (posterior probabilities) for Fig. 2B and 2C must be provided.

Additional comments

Line 93. It is important that the authors report the details of the analysis
as they performed it, and not simply refer to King et al. 2017.

Line 104. Mention the Github repository specifically.

The names of the formations from the early Devonian of Spitzbergen in Fig. 1C are not mentioned in the text. Are these the same as MOTH and Turni Hill? It doesn't seem so; there seems to be a mistake here.

Given that Fig. 1C is a histogram and not a probability density function, I don't think it's appropriate to refer to the y-axis as "density."

Figs. 2B and 2C need to have scale bars. Also, B and C are not "cladograms" despite what the
figure legend says. They are chronograms, and support values should also be supplied.

Line 142: "The 50% majority-rule consensus tree for the analysis...". The authors do not state how the trees in Fig. 2B and 2C were produced. Typically, BEAST users use the TreeAnnotator program to produce a maximum clade credibility tree (MCCT). This is not the same as at 50% maj rule, which cannot be produced by Tree Annotator. Please state how
the maj-rule tree was produced, and why should it be preferred over the MCCT.

Line 148: "The results show that linking tip-ages from fossil sites can lead to an increase in the precision of age estimates when compared with independent tip dates"

Line 170: "Linking tip-dates can affect the phylogenetic position of fossils. The affected taxon, Cassidiceps, from the MOTH site, has the oldest age estimate...". The authors argue that the change in the position of Cassidiceps between Fig. 2B and 2C is a significant effect of linking tip-dates. In truth, the difference is minimal, and without branch support values (posterior probabilities), one cannot really assess the importance of the difference.

Fig. 2A. The increased precision that is claimed to result from linking the tip dates is not that striking. Although I think the method in general is important, the actual results in this analysis aren't very convincing that linked tip priors have an important effect, in this case. Nonetheless, I think what the authors propose is an important advance.

·

Basic reporting

The manuscript is well-written and easy to follow along. I only have very minor comments on my reviewer PDF. One thing I think would help out is an additional context figure that compares the stratigraphy/rock formations/ages of the two sites, just to help visualize the issue at hand with the example case. Otherwise, the manuscript is concise and self-contained as they explore their new protocols.

Experimental design

I feel the design was straightforward in the implementation of their new protocols. One thing to consider, is maybe include a second example data set, perhaps one with a wider temporal range on the order of tens of millions of years as the current example is narrower, <10 millions of years.

Validity of the findings

I think the linking of sites and taking stratigraphic ordering into account is a good idea that would be applicable for many researchers that utilize tip-dating methods.

Additional comments

This was a concise read and proportional to what was being explored. I only have minor comments and I look forward to the final publication.

---

## Round 0.2 · accepted · Accept

Dear authors,

It is a pleasure to accept your manuscript entitled "Tip dating with fossil sites and stratigraphic sequences" which you submitted to PeerJ.
Once again, thank you for submitting your manuscript to PeerJ.

Best wishes,

Raquel López-Antoñanzas

Reviewer 2 ·

Basic reporting

No comment

Experimental design

No comment

Validity of the findings

No comment